# Antibiogram of clinical isolates from primary and secondary healthcare facilities: A step towards antimicrobial stewardship

**Isabel Naomi Aika** [ID]*, **Ehijie Enato**

Department of Clinical Pharmacy & Pharmacy Practice, Faculty of Pharmacy, University of Benin, Benin, Nigeria

* Isabel.aika@uniben.edu

**Data Availability Statement:** The necessary supporting data for this research has been made available in the submission.

## Abstract

Antibiogram development and use is a core element of antimicrobial stewardship practice, such data is scarce in healthcare settings in developing countries. The study aims to determine the epidemiology of clinical isolates and their antibiograms in secondary healthcare (SHC) and primary healthcare (PHC) facilities in Benin City, Nigeria. This was a retrospective study in three laboratories in SHC and PHC facilities. Microbial culture and susceptibility report over the past 4 years was collated. Ethical Clearance was obtained from the Edo State Ministry of Health and Benin City. Data were entered and analyzed using SPSS version 22. Of the 819 isolates, urine, semen and vagina swab were most represented (50.7%; 16.1%; 13.2%). *S.aureus* (60.6%) and coliform organism (31.9%) were commonly isolated. High resistance of 75%->90% was seen with penicillin, cephalosporin, macrolide, aminoglycoside and floroquinolone against *S. aureus*, *Ps aeruginosa*, coliform and *E. coli*. Resistance to all antimicrobials was observed in 11.7% of the isolates, multidrug resistance (MDR) was found to be 61.4%. MDR for *Ps aeruginosa*, coliform, E coli, and *staph aureus* were 95.5%, 67.3% 25.6% and 82.8% respectively. *In the SHC, 15.4% of isolates were resistant to all antibacterial compared to none in the PHC. There was consistent yearly increase in resistance to more than six agents in both centers. Gender difference in antimicrobial resistance was observed*. High MDR observed in this study emphasizes the need for routine antibiogram and its use in updating treatment guidelines to reflect the current resistance pattern to available antimicrobials.

## Introduction

The discovery and development of antimicrobial agents has significantly changed the negative narrative of how infectious diseases has plagued the human race over history. However, due to slow pace at which these agents are developed and the emergence of resistance to antimicrobials, treating common pathogens has become a challenge. Different types of resistance have developed over time resulting in their classification as multidrug resistance MDR, extended resistance (XDR) and Pandrug resistance (PDR) [1].

**Funding:** The authors received no specific funding for this work.

**Competing interests:** The authors have declared that no competing interests exist.

There is an increasing prevalence of pathogenic multidrug-resistant bacteria globally. Infections with multidrug-resistant bacteria are hard to treat since fewer or no treatment options remain. In some cases, health care providers have to use antibiotics that are more toxic for the patient resulting in increased duration of illness and prolonged hospital stay, excess costs, number of complications, and deaths. Data from U.S intensive care unit studies and surveillance studies estimate that by 2050, 10 million deaths are projected to occur due to antimicrobial resistance (AMR), while another study projected AMR to cost the global economy US$100 trillion, in the same period [2, 3]. Against this backdrop, the World Health Organization (WHO) On 27 February 2017, published the list of global priority pathogens (GPP)–a catalog developed through a consultative process [4, 5]. The most critical organisms in this catalog includes multidrug resistant bacteria that pose a particular threat in hospitals, nursing homes, and among patients whose care requires devices such as ventilators and blood catheters. They include *Acinetobacter*, *Pseudomonas* and various Enterobacteriaceae (including *Klebsiella*, *E. coli*, *Serratia*, and *Proteus*) [4].

The World Health Organization (WHO) emphasizes the key role of the microbiology laboratory in antimicrobial stewardship (AMS) by informing the appropriate use of antibiotics through development of antibiograms [6]. Comprehensive data on the antibiogram of bacterial pathogens isolated from different body sites of infections is needed for surveillance systems which aid in monitoring antimicrobial use and resistance thus improving decision making and assessing the effect of interventions at the local, national and international level. There is scarcity of such data in developing countries including Nigeria [7]. Networking between laboratories may increase infection surveillance within a huge geographical region. The WHO supports the establishment of national networks for the regular exchange of information and proper support for the laboratory. However, only a few countries have local, national and international laboratory networks. The establishment of an integrated laboratory system is very important to combat many infectious diseases. In Low- Middle- Income- Countries (LMICs), there are few or no documented data on the bacterial isolates and antibiogram profiles in healthcare facilities even at local level and integrated systems are financially neglected in developing countries [8]. Therefore, this study aims to determine the epidemiology of microbial isolates, to appraise and compare the antibiogram and resistogram in secondary and primary healthcare facilities in Benin City, Nigeria.

## Materials and methods

### Study design/Setting

This was a Cross-sectional retrospective study conducted in Edo State, located in Southern Nigeria. The study was carried in one laboratory in a secondary healthcare facility and two laboratories located in two local government headquarters associated with primary healthcare centers (PHC).

Ethics Statement: Ethical approval was obtained from the department of Medical Services of Edo State Ministry of Health (Reference No- HA-737/45). Administrative approval was sought from all facilities included in the study. The requirement of obtaining patient's informed consent before accessing their medical records from the microbiological laboratories for this study was waived. All data collected were kept confidential.

### Data collection

A retrospective review of microbial culture and antibiotic susceptibility entries in the microbiology unit of the Laboratory Services in PHC and SHC facilities between April 2018 to May 2021 was performed. The microbiology unit in the SHC receives specimens from inpatient

and outpatient departments, while that in the PHC receives outpatient samples for bacteriologicl investigation. The bacteriological investigation involves culturing of the specimen with appropriate culture media in accordance with the guideline of Clinical and Laboratory Standards Institute (CLSI) [9]. Bacterial isolates were identified by standard phenotypic microbiological methods, and the Antibiotic Susceptibility testing of isolates was done by Kirby-Bauer disk diffusion on Muller Hinton Agar (Oxoid UK) [9]. The following antibiotic disks were used: Amoxicillin-Clavulanate, Ceftriaxone, Ofloxacin, Ciprofloxacin, Nitrofuratoin, Ampicillin, Cefuroxime, Ceftazidime, Cefixime, erythromycin, Gentamicin, Cloxacillin for the secondary facility; Amoxicillin-Clavulanate (Amox-Clav), Ceftriaxone, Ofloxacin, Ciprofloxacin, Nitrofuratoin, Cefuroxime, Ceftazidime, Cefixime, Erythromycin, Gentamicin, Amoxicillin, Levofloxacin, Azithromycin, Cloxacillin and Streptomycin for the primary facility. The choice of antibiotics arises from commonly prescribed drugs from physicians in the region. Quality control was done using reference strains such as *Staphylococcus aureus* (ATCC25923), *E. Coli* (ATCC 25922), *Pseudomonas aeruginosa* (ATCC 27853). Results were interpreted based on sizes of zone of inhibition from the CLSI guideline [9]. All entries with patient's age, sex, source of specimen, type of organism isolated and antimicrobial sensitivity test results were included in the study (S1 Text). This study excludes entries lacking any of the information mentioned above, and those with fungi growth such as candida albicans or other microorganisms that are not bacteria.

## Data analysis

Data were entered and analyzed using SPSS version 22. Proportion of specimen and microorganisms isolated from each facility were expressed as percentages. The relationship between gender of patients versus susceptibility of isolates and respective facility and susceptibility tests were compared using the Chi square ($\chi^2$) test. Statistical differences were considered significant at p value of <0.05. the pattern of resistance over the study period was analyzed in both facilities and individual facility.

## Results

Eight hundred and nineteen clinical isolates subjected to culture and sensitivity test was used for this study. Age range of patients' samples was between 4 months to 90 years, of which 20 (2.3%) were <18 years, 16 (1.95%) were ≤ 12 years, and 799 (97.5%) were adults that is ≥ 18years. Three hundred and ninety five of the clinical isolates were from males, while and 424 (51.8%) were from females. Majority (75.8%) of the samples were retrieved from secondary institution. In 2018, 4.6% of the total of isolates were reported from PHC only. In 2019, 32.6% of total isolates from both centers were represented, about half (46.9%) and15.9% of the isolates were reported in the year 2020 and 2021 from both centers respectively.

Urine (50.7%) sample represented half of all samples followed by semen (16.1%) and vagina swab (13.2). Urine (51.8%), semen (16.5%), wound swab (13.3%) and vagina swab (6.9%) represented majority of the specimen in SHC, while urine (46.9%), vaginal swab (32.8%) and semen (14.6%) were most encountered in PHC. *S.aureus* (60.6%) was the microorganism most isolated while coliform organism (31.9%) represented about one-third of the microorganisms. *Ps aeruginosa* and proteus were isolated from SHC only. The only Gonorrhea isolate was seen from PHC while the PHC differentiated *E.coli* from other coliforms unlike in SHC where all are grouped together (See Table 1). About half (51.8%) of the isolates were collected from females with urine (54.7%), wound swab (12.3%) and vagina swab (25.5%) constituting most of the samples. In males, most samples were from urine (46.3%), semen (33.4%) and wound swab (8.8%). Coliform organisms and *S.aureus* constituted majority of the organisms isolated

**Table 1. Specimen and microorganisms isolated N = 819.**

| Variable | [N(%)] | SHC [N(%)] | PHC [N(%)] |
|---|---|---|---|
| **Specimen** | | | |
| Urine | 415 (50.7) | 322 (51.8) | 93 (46.9) |
| Blood | 7 (0.9) | 5 (0.8) | 2 (1) |
| Semen | 132 (16.1) | 103 (16.5) | 29 (14.6) |
| Ear swab | 21 (2.6) | 20 (3.2) | 1 (0.5) |
| Wound swab | 87 (10.6) | 83 (13.3) | 4 (2) |
| Vagina swab | 108 (13.2) | 43 (6.9) | 65 (32.8) |
| *Others | 49 (6.0) | 45 (7.2) | 4 (2) |
| Total | 819 | 621 | 198 |
| **Microorganisms** | | | |
| *Ps Aeruginosa* | 22 (2.2) | 22 (100) | 0 |
| *Coliform* | 261 (31.9) | 257 (99) | 4 (0.1) |
| *S.aureus* | 496 (60.6) | 336 (67.7) | 160 (22.3) |
| *E.coli* | 32(3.9) | 1 (3) | 31 (97) |
| *Gonorrhea* | 1(0.1) | 0 | 1 (100) |
| *Strept Pneumonia* | 5(0.6) | 3 (60) | 2 (40) |
| *Proteus* | 2(0.2) | 2 (100) | 0 |

*Includes urethral swab, Synovial fluid, penile swab, ovarian aspirate

from all cultures with highest isolates of *S.aureus* from urine (45.7%), semen (21.7%), vagina swab (16.3%); for coliform, urine (44.4%) and wound swab (16.8%). *Ps Aeruginosa* was more isolated from wound swab (22.7%) and semen (27.2%). Most E. coli isolated was from urine (56.2%) and vagina swab (31.2%), all streptococci pneumonia was isolated from urine, proteus specie was isolated from urine (50%) and ear swab (50%) respectively, while the only *N. Gonorrhea* was found in vagina swab.

Table 2 describes the number of antimicrobials isolates are sensitive to in accord with the individual healthcare institution. Only one isolate is sensitive to all antimicrobials. From the PHC, majority of the isolates are sensitive to between 3 to 5 antimicrobials, while in the SHC, majority of the organisms are sensitive to between 1 to 3 antimicrobials. More resistance was seen in the SHC as 15.4% of isolates were resistant to all antimicrobials. Of the 819 microorganisms, at least 61.4% are multidrug resistant (MDR). Multidrug resistance of specific

**Table 2. Isolate's sensitivity to number of antimicrobials isolates are Sen.**

| Number of Antimicrobial | SHC Isolates N (%) | PHC Isolates N (%) | TOTAL N (%) |
|---|---|---|---|
| None | 96 (15.4) | 0 | **96 (11.7*)** |
| 1 | 208(33.5) | 1(0.5) | **209 (25.5*)** |
| 2 | 151(24.3) | 8(4) | **159 (19.4*)** |
| 3 | 101(16.3) | 20(10) | **121 (14.8*)** |
| 4 | 44(7.1) | 79(40) | 123 (15) |
| 5 | 12(1.9) | 67(34) | 79 (9.6) |
| 6 | 6(0.9) | 22(11) | 28 (3.4) |
| 7 | 2(0.3) | 1(0.5) | 3 (0.4) |
| 8 | 1(0.2) | 0 | 1 (0.1) |
| Total N (%) | 621(75.8) | 1. (24.2) | 819(100) |

*MDR = 61.4%

microorganisms was noted, for *ps. Aeruginosa* (95.5%), coliform (67.3%), E.coli (25.6%), and *S.aureus* (82.6%).

Seven antimicrobials (amox-clav, ampicilin, ceftazidime, cefixime, cloxacillin, streptomycin and amoxycillin) showed more than 90% resistance to majority of the isolates. Ciprofloxacin, cefuroxime and erythromycin showed more than 75% resistance. *N. Gonorrhae* was only sensitive to Cefixime, amox-clav and azithromycin. Proteus was resistant to all antimicrobial agent, while *Ps. Aureginosa* showed more sensitivity to ciprofloxacin and ofloxacin but high resistance to other antimicrobials. Coliform was highly sensitive to ciprofloxacin (96%) but showed less than 40% sensitivity to other antimicrobials. *S aureus* showed the highest sensitivity to levofloxacin (62.6%) (See Table 3).

Isolates from males and females were more sensitive to gentamicin (30% versus 39%), ofloxacin (34% versus 38%) and ceftriaxone (35% versus 39%) compared to other antimicrobial

**Table 3. Microbial sensitivity pattern of isolates (N = 819).**

| Antimicrobial | | Ps Aeruginosa N (%) | Coliform N (%) | S.aureus N (%) | E.coli N (%) | N. Gonorrhea N (%) | Strep Pn N (%) | Proteus N (%) | Total N (%) |
|---|---|---|---|---|---|---|---|---|---|
| Amox-Clav | R | 21(95.4) | 247(92.5) | 454(96) | 31(97) | 0 | 4(80) | 2(100) | 759(92.6) |
| | S | 1 (5.6) | 14(7.5) | 42(4) | 1(3) | 1(100) | 1(20) | 0 | 60(7.4) |
| Ceftriaxone | R | 18(82) | 211(79) | 269(54) | 5(15.6) | 1(100) | 4(80) | 2(100) | 510(62.3) |
| | S | 4(18) | 50(21) | 227(46) | 27(84.4) | 0 | 1(20) | 0 | 309(37.7) |
| Ofloxacin | R | 12(54.5) | 158(60.5) | 326(66) | 19(59) | 1(100) | 5(100) | 1(100) | 522(63.7) |
| | S | 10(45.5) | 103(39.5) | 170(34) | 13(41) | 0 | 0 | 0 | 297(36.3) |
| Cipro | R | 0 | 2(4) | 157(82) | 28(87.5) | 1(100) | 2(100) | 0 | 535(85) |
| | S | 2(100) | 54(96) | 35(18) | 3(12.5) | 0 | 0 | 0 | 94(15) |
| Nitro | R | 20(91) | 147(57) | 258(76) | 1(25) | 0 | 1(33.3) | 2(100) | 429(68.1) |
| | S | 2(9) | 112(43) | 82(24) | 3(75) | 0 | 2(66.7) | 0 | 201(31.9) |
| Ampicillin | R | 22(100) | 258(99.6) | 335(99) | 1(100) | 0 | 3(100) | 2(100) | 621(95.2) |
| | S | 0 | 1(0.4) | 3(1) | 0 | 0 | 0 | 0 | 4(4.8) |
| Cefuroxime | R | 19(83.6) | 226(87) | 362(73) | 27(84.4) | 1(100) | 5(100) | 2(100) | 642(78.4) |
| | S | 3(16.4) | 35(13) | 134(27) | 5(15.6) | 0 | 0 | 0 | 177(21.6) |
| Ceftaz | R | 21(95.4) | 231(89) | 331(98) | 31(97) | 1(100) | 2(40) | 0 | 588(94.1) |
| | S | 1(5.6) | 28(11) | 7(2) | 0 | 0 | 3(60) | 2(100) | 37(5.9) |
| Cefixime | R | 22(100) | 244(91) | 473(95.4) | 25(78) | 0 | 5(100) | 2(100) | 771(94.1) |
| | S | 0 | 17(8) | 23(4.6) | 7(22) | 1(100) | 0 | 0 | 48(5.9) |
| Erythromy | R | 21(95.4) | 251(96) | 411(83) | 24(75) | 1(100) | 4(80) | 2(100) | 714(87.2) |
| | S | 1(5.6) | 10(4) | 85(7) | 8(25) | 0 | 1(20) | 0 | 105(12.8) |
| Gentamicin | R | 13(52) | 176(67.4) | 316(64) | 18(56) | 1(100) | 3(60) | 2(100) | 529(64.6) |
| | S | 9(48) | 85(32.6) | 180(36) | 14(44) | 0 | 2(40) | 0 | 290(35.4) |
| Cloxacillin | R | 22(100) | 256(98) | 332(98) | 1 | 0 | 3(100) | 2(100) | 616(98.4) |
| | S | 0 | 3(2) | 7(2) | 0 | 0 | 0 | 0 | 10(1.6) |
| Levo | R | 0 | 1(50) | 59(37.4) | 12(40) | 1(100) | 2(100) | 0 | 75(38.9) |
| | S | 0 | 1(50) | 99(62.6) | 16(60) | 0 | 0 | 0 | 118(61.1) |
| Azithro | R | 0 | 0 | 66(58) | 16(53) | 0 | 0 | 0 | 82(42.5) |
| | S | 0 | 2(100) | 92(42) | 14(42) | 1(100) | 2(100) | 0 | 111(57.5) |
| Amoxy | R | 0 | 2(100) | 155(98) | 30(97) | 1(100) | 1(100) | 0 | 189(97.9) |
| | S | 0 | 0 | 3(2) | 1(3) | 0 | 0 | 0 | 4(2.1) |
| Clindamy | R | 0 | 1(50) | 66(58) | 12(39) | 1(100) | 0 | 0 | 80(41.5) |
| | S | 0 | 1(50) | 92(42) | 19(61) | 0 | 1(100) | 0 | 113(58.5) |
| Strepto | R | 0 | 2(100) | 155(98) | 30(97) | 1(100) | 1(50) | 0 | 189(98) |
| | S | 0 | 0 | 3(2) | 1(3) | 0 | 1(50) | 0 | 5(2) |

**Table 4. Relationship between sex and antimicrobial sensitivity.**

| Antimicrobial | | Male N (%) | Female N (%) | Total N (%) | P-Value |
|---|---|---|---|---|---|
| Amox-Clav | R | 358(47) | 401(53) | 759(92.6) | 0.021* |
| | S | 37(62) | 23(38) | 60(7.4) | |
| Ceftriaxone | R | 254(49.8) | 256(50.2) | 510(62.3) | 0.139 |
| | S | 141(45.6) | 168(54.4) | 309(37.7) | |
| Ofloxacin | R | 260(49.8) | 262(50.2) | 522(63.7) | 0.130 |
| | S | 135(45) | 162(55) | 297(36.3) | |
| Ciprofloxacin | R | 282(53) | 253(47) | 535(85) | 0.002* |
| | S | 36(38) | 58(62) | 94(15) | |
| Nitro | R | 245(57) | 184(43) | 429(68.1) | 0.000* |
| | S | 74(37) | 127(63) | 201(31.9) | |
| Cefuroxime | R | 318(49.5) | 324(40.5) | 642(78.4) | 0.091 |
| | S | 77(43.5) | 100(56.5) | 177(21.6) | |
| Ceftazidime | R | 299(51) | 289(49) | 588(94.1) | 0.047* |
| | S | 17(46) | 20(54) | 37(5.9) | |
| Cefixime | R | 370(48) | 401(52) | 771(94.1) | 0.344 |
| | S | 25(52) | 23(48) | 48(5.9) | |
| Erythromy | R | 333(47) | 381(53) | 714(87.2) | 0.012* |
| | S | 62(59) | 43(41) | 105(12.8) | |
| Gentamicin | R | 273(52) | 256(48) | 529(64.6) | 0.006* |
| | S | 122(42) | 168(58) | 290(35.4) | |
| Cloxacillin | R | 311(50.4) | 305(49.6) | 616(98.4) | 0.013 |
| | S | 7(70) | 3(30) | 10(1.6) | |
| Levofloxacin | R | 32(43) | 43(57) | 75(38.9) | 0.063 |
| | S | 47(40) | 71(60) | 118(61.1) | |
| Azithromycin | R | 28(34) | 54(66) | 82(42.5) | 0.018* |
| | S | 51(46) | 60(54) | 111(57.5) | |
| Amoxycillin | R | 78(41) | 111(59) | 189(97.9) | 0.042 |
| | S | 1(25) | 3(75) | 4(2.1) | |
| Clindamycin | R | 36(45) | 44(55) | 80(41.5) | 0.043* |
| | S | 43(38) | 70(62) | 113(58.5) | |
| Streptomycin | R | 79(42) | 110(58) | 189(98) | 0.006 |
| | S | 0 | 5(100) | 5(2) | |

agents. In addition, isolates from females showed sensitivity to nitrofurantoin (30%). Table 4 describes the relationship between gender and sensitivity pattern to antimicrobials. Isolates from males were more sensitive to amox-clav (p-0.021), erythromycin (p-0.012) and azithromycin (p-0.018) compared to females, while females showed more sensitivity compared to males to ciprofloxacin (p-0.002), nitrofuratoin (p-0.000), ceftazidime (p-0.047), gentamicin (p-0.006) and clindamycin (p-0.043). All isolates from males were resistant to streptomycin compared to those from females, similar resistance can be observed for cefixime, ofloxacin and ceftriaxone which is about 50% for either gender. Of the isolates sensitive to levofloxacin, 60% were from females, although there was no statistical significance compared to isolates from males.

Isolates from SHC had the highest sensitivity to gentamincin (30%) and nitrofurantoin (31%), while those from PHC showed highest sensitivity to ceftriaxone (86.4%), gentamicin (47.5%) and cefuroxime (30%). It is observed that few isolates were used to test for sensitivity with some antibiotics such as streptomycin, ciprofloxacin, ampicillin and ceftazidime in both facilities due to differences in antibiotic discs used from both centers, and at some point, the

**Table 5. Relationship between antimicrobial sensitivity pattern and healthcare institution.**

| Antimicrobial | | SHC N (%) | PHC N (%) | Total N (%) | P-Value |
|---|---|---|---|---|---|
| Amox-Clav | R | 566(91) | 193(97.5) | 759(92.6) | 0.0001* |
| | S | 55(9) | 5(2.5) | 60(7.4) | |
| Ceftriaxone | R | 483(77.7) | 27(13.6) | 510(62.3) | 0.000* |
| | S | 138(22.3) | 171(86.4) | 309(37.7) | |
| Ofloxacin | R | 417(67) | 105(53) | 522(63.7) | 0.000* |
| | S | 204(33) | 93(47) | 297(36.3) | |
| Ciprofloxacin | R | 533(86) | 2(25) | 535(85) | |
| | S | 88(14) | 6(75) | 94(15) | |
| Nitrofurantoin | R | 427(69) | 2(78) | 429(68.1) | |
| | S | 194(31) | 7(22) | 201(31.9) | |
| Ampicillin | R | 617(99) | 7(100) | 621(95.2) | |
| | S | 4(1) | 0 | 4(4.8) | |
| Cefuroxime | R | 504(81) | 138(70) | 642(78.4) | 0.01* |
| | S | 117(19) | 60(30) | 177(21.6) | |
| Ceftazidime | R | 584(94) | 4(100) | 588(94.1) | |
| | S | 37(6) | 0 | 37(5.9) | |
| Cefixime | R | 594(96) | 177(89.4) | 771(94.1) | 0.02* |
| | S | 27(4) | 21(10.6) | 48(5.9) | |
| Erythromycin | R | 568(91) | 146(74) | 714(87.2) | 0.006* |
| | S | 53(9) | 52(26) | 105(12.8) | |
| Gentamicin | R | 435(70) | 94(52.5) | 529(64.6) | 0.00* |
| | S | 186(30) | 104(47.5) | 290(35.4) | |
| Cloxacillin | R | 613(98.7) | 3(60) | 616(98.4) | |
| | S | 8(1.3) | 2(40) | 10(1.6) | |

facility also changed the discs used for sensitivity testing. Table 5 shows that isolates from PHC were more resistant to augmentin (p<0.05) compared to those from SHC, while isolates from SHC were more resistant to ceftriaxone, ofloxacin, cefuroxime, cefixime, erythromycin and gentamicin (p<0.05) than isolates from PHC.

The sensitivity patterns were analyzed in both PHC and SHC over the study period, and the resistance profiles were observed. Resistogram shows that microorganisms have been consistently resistant (>60%) to Ciprofloxacin, Amox-Clav, Nitrofurantoin, Ampicillin, Ceftazidime, Erythromycin, Cloxacillin and Gentamicin in SHC over the past three years. There was a rise in the resistance profiles of cefuroxime and ceftriaxone between 2019 and 2020, followed by a drop in resistance between the year 2020 and 2021 (Fig 1). Similarly, in the PHC over the past 4 years, resistance to Amox-Clav, Cefixime, Streptomycin and Amoxicillin was noted (Fig 2), resistance to Azithromycin rose from 30% in 2018 to 74% in 2021. Resistance to Ceftriaxone seems to be increasing after a 2 years drop in resistance. Resistance to gentamicin was observed to be fluctuating with alternate year of increase resistance. There has been a steady decrease in resistance to levofloxacin over the study period. Clindamycin saw a drop in resistance from 49% to 40% in between 2020 and 2021. Similar pattern of resistance were observed for Cefixime, Amox-Clav, Ofloxacin and Gentamicin in both PHC and SHC over the years.

## Discussion

This study sought to determine and compare the antimicrobial sensitivity pattern over time in PHC and SHC facilities. Urine, semen and vagina swab were the most common samples

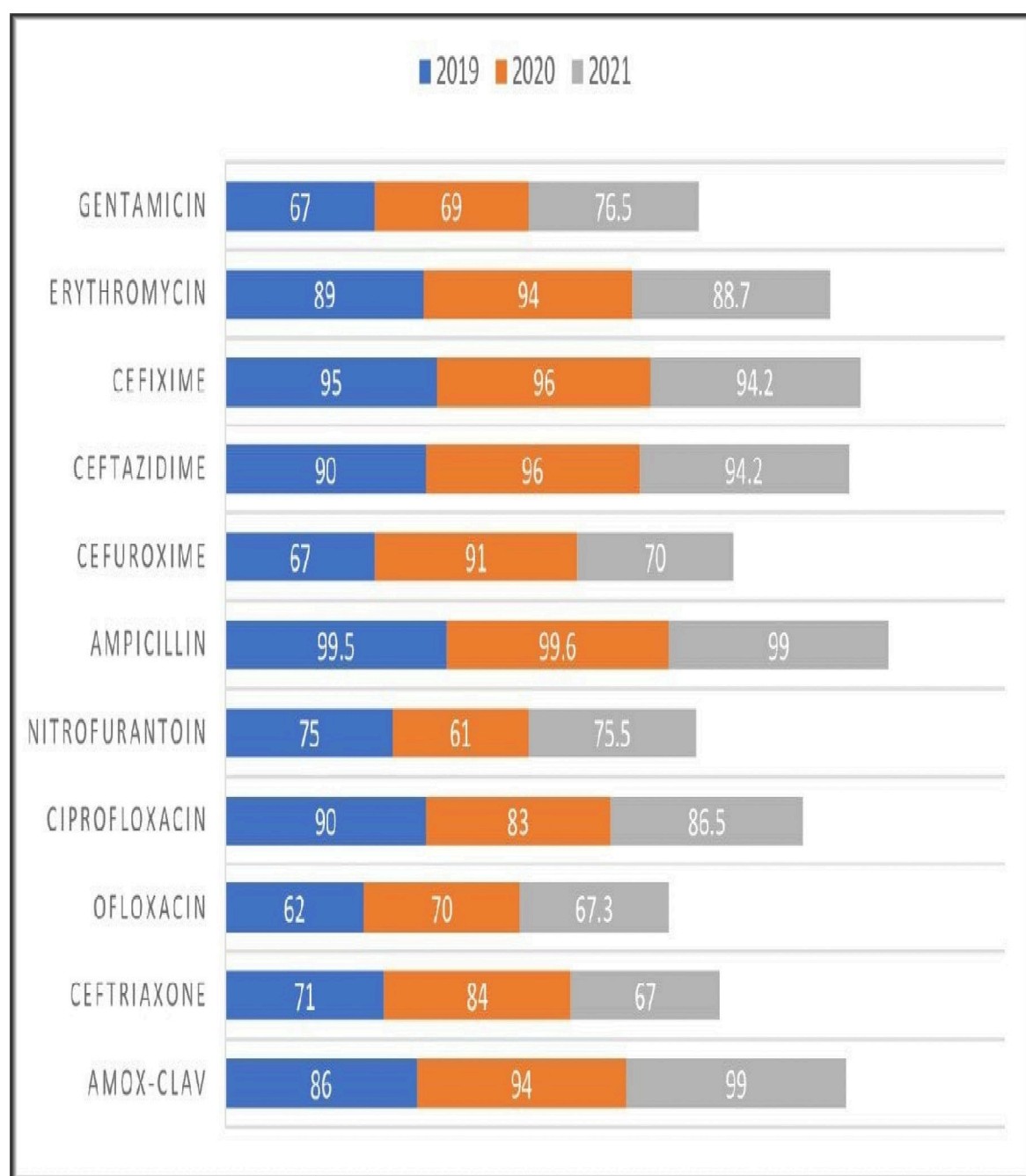

**Fig 1. Antimicrobial resistance pattern in SHC over 3 years (%).**

isolates were collected for microbiology investigation over the study periods from both facilities. This suggests that urinary tract infections (UTIs) and infections of the reproductive organs were common reasons for ordering microbial culture and sensitivity tests in these facilities. One reason for this observation is that respiratory samples and cerebrospinal fluid are more associated with paediatric patients and physicians in these facilities may tend to adopt presumptive diagnosis in treatment of these patients, and considering the fact that viruses cause respiratory infections more than bacteria. Moreso, investigation with CSF requires more

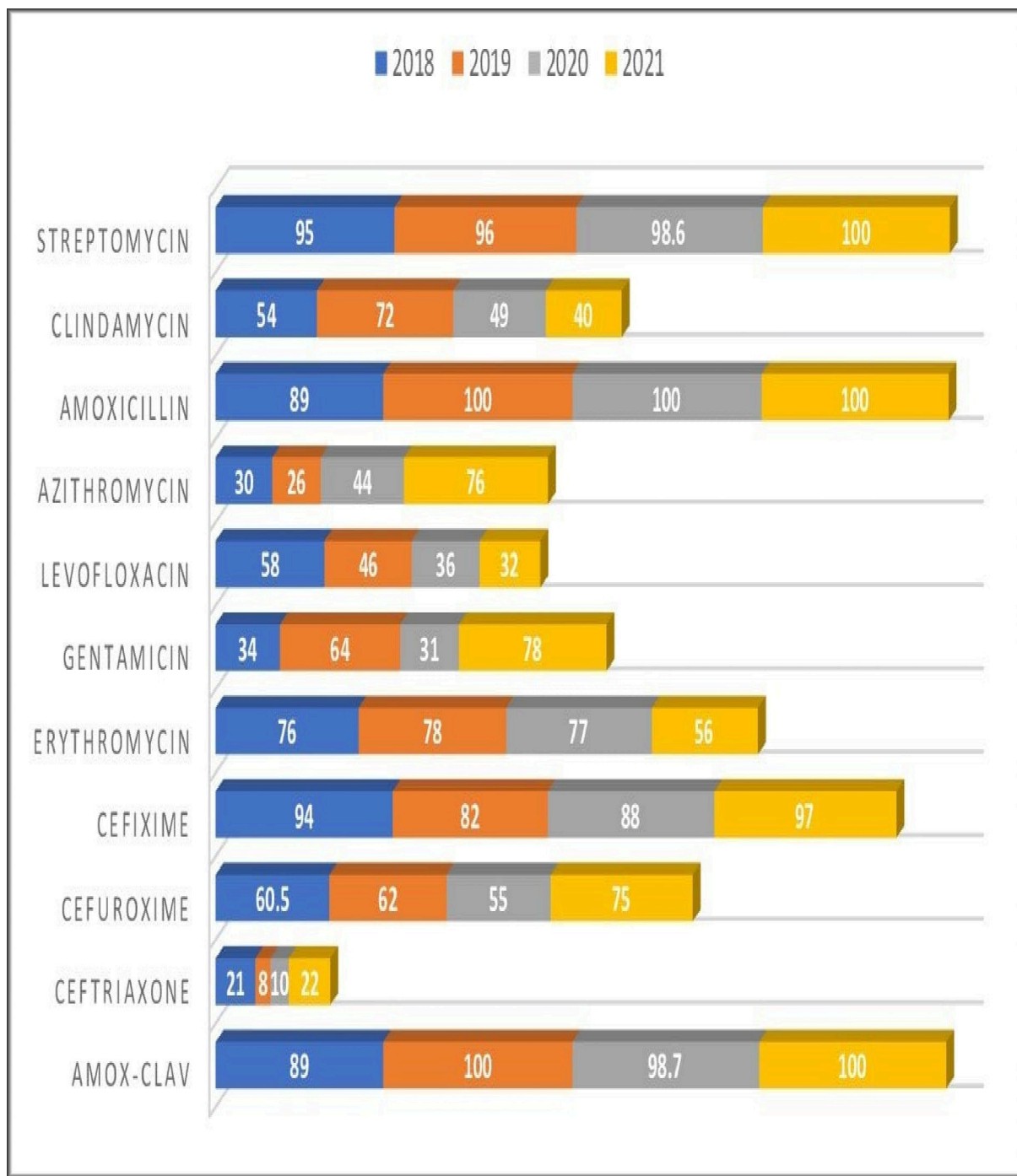

**Fig 2. Antimicrobial resistance pattern over 4 years in PHC (%).**

expertise which may be lacking in primary care settings Another possibility is that some patients might have treated infections associated with urinary tract and the vagina on their own from patent medicine dealers and pharmacy outlets most of which do not ask patients to conduct microbial test before treatment as is common in LMICS, hence recurrent infections warrant them to visit these facilities [6]. Urine sample made up half of all culture, majority of

which was from females. This could be because UTIs occur more frequently in females than in males at a ratio of 8:1, approximately 50–60% of women will report at least one UTI in their lifetime [10]. *S.aureus* was most isolated from semen, urine and vagina swab followed by E.coli from vaginal swab. More evidence is currently emerging to show that Staphylococcus, particularly Staphylococcus aureus, can colonize the reproductive systems and affect their structure and function. In a study of a total of 140 sperm samples collected from the University of Benin Teaching Hospital, *S. aureus* (28.3%) was the most common pathogens [11], in a related study a prevalence rate of 38.7% for *S. aureus* from high vaginal swab and endocervical swabs and a prevalence of 75% from semen cultures of infertile couples was reported [12]. Another investigation identified *S. aureus* as the most prevalent vaginal pathogen (57.33%) among local infertile women, followed by *Escherichia coli* (25.33%) [13].

The relationship between *S. aureus* and E. coli causing urinary tract and reproductive infections is linked with the fact that approximately 30% of the human population is colonized with these microbe, and E.coli is a common contaminate of drinking water and the environment [14]. Transmission of urinary tract infections into the reproductive system is unavoidable. Moreover, a range of 50–80% of UTI patients are infected repeatedly by the same bacterial strain, indicating the presence of reservoirs within the host [15]. Few studies from other parts of the world show that *Staph aureus* is a cause of UTI or vaginal/prostatic infections, our study however reveals increased infections associated with this organism. More studies should give attention to this finding for possible inclusion in guidelines for treatment.

Results from this study show that most of the coliforms isolated were from the secondary health care facility, whereas in the PHC, E.coli was clearly differentiated. It is possible that the method of identification and reporting is adopted either out of convenience or limitations in laboratory differential of this group of organisms. Ideally identification and reporting as E. coli and non- E coli coliforms or further characterization of the non-E coli coliform as *Klebsiella or Citrobacter* can help the clinician to make appropriate decision when interpreting the microbial sensitivity results, this is important as some coliform like *Klebsiella* and E coli are classified among the WHO GPP [4, 16].

## Sensitivity pattern of microbial isolates

In this study, high resistance was seen with penicillin, cephalosporin, macrolide, aminoglycoside and some floroquinolone antimicrobials against *S. aureus*, *Ps aeruginosa*, coliform and E. coli. cephalosporin and penicillin had the highest resistance of greater than 90%, erythromycin and ciprofloxacin had resistance of 75%, levofloxacin and azithromycin had sensitivity of 61.1% and 57.5% respectively, these later antibacterial were used as part of sensitivity testing in PHC facility. Resistance to all antimicrobials was observed in 11.7% of the isolates in this study, multidrug resistance (MDR) of isolate was found to be 61.4% defined as resistance to all antimicrobials or resistance to 3 or more classes of antimicrobial agents [1]. For specific organisms, MDR for *Ps aeruginosa*, coliform, E coli, and staph aureus were 95.5%, 67.3% 25.6% and 82.8% respectively. *Ps Aerugenosa* was more sensitive to floroquinolone (ciprofloxacin and ofloxacin) than to other agents. High MDR has been reported in western Nigeria, where 59.3% of bacterial isolates were MDR strains, while in Ethiopia, 61.8% MDR was reported [17, 18]. Multidrug resistance for *Ps aeruginosa* and *S aureus* are quite high when compared with other studies, for S aureus 60.9% was reported in Ethiopia, and MDR for E. coli is higher than in this study (50%) and this could be because the E.coli component of coliform organisms isolated in this study was not differentiated [17]. In Ibadan Nigeria, MDR among *Pseudomonas* isolates was 60%, these isolates were resistant to almost all antimicrobials including floroquinolones and against Amikacin and Meropenem [18, 19].

## Comparison of antimicrobial resistance pattern and gender

Isolates from females were more sensitive to ciprofloxacin, nitrofurantoin, ceftazidime and gentamicin than those from males, while isolates from males were more sensitive to amox-clav, erythromycin and azithromycin compared to isolates from females. Studies are scarce on the relationship between gender and antimicrobial sensitivity. In a study conducted in Bangladesh on *E. coli* isolates from urine, the authors reported that nitrofurantoin and colistin were less effective against isolates obtained from males compared to isolates obtained from females [19]. In another study, age difference in antimicrobial sensitivity has been noted among paediatrics and adult population, while *g*ender difference in bacterial infections and prescription has been observed where women are 27% more likely to receive a prescription for antibiotics than men and women are more likely than men to be prescribed cephalosporins and macrolides during their lifetimes [20, 21]. One reason for gender differences in bacterial infections has been connected to genetic background. More investigation should be considered to determine the relevance of gender-based differences in antimicrobial resistance [22].

## Comparison of antimicrobial resistance pattern in PHC and SHC

In the SHC, 15.4% of isolates were resistant to all antibacterial compared to none in the PHC while isolates from SHC were sensitive to between 1–3 antibacterial agent, but in the PHC, in general terms, isolates from SHC are more resistant to those from PHC. Isolates from PHC are more resistant to amox-clav compared with isolates from SHC facilities, this could likely be as a result of overprescribing and use of amox-clav both as a self-medication and in primary health care centers. Although both centers showed resistance to ciprofloxacin, ofloxacin, amox-clav and gentamicin, isolates from SHC showed more resistance to ceftriaxone, ofloxacin, cefuroxime, cefixime, erythromycin and gentamicin compared to isolates from PHC. Few studies are available to compare antimicrobial sensitivity patterns at different healthcare levels. A close study in India compared antimicrobial resistance between urban and rural community settings. The authors observed a higher resistance to some antimicrobial agents in rural sites than in urban sites and vice versa. Unexpectedly, *Staphylococcus aureus* resistance was higher in the rural site compared to the countries' resistance surveillance profile for ciprofloxacin and clindamycin. The authors concluded that the burden of AMR high in both rural and urban sites, reinforcing that AMR burden cannot be ignored in rural settings [23]. Over the past 4 years of study, there was consistent increase in resistance to amox-clav, ceftriaxone, streptomycin, amoxicillin and surprisingly azithromycin in PHC, while in the SHC, increase in ciprofloxacin, amox-clav, nitrofurantoin, ampicillin, ceftazidime, erythromycin, cloxacillin and gentamicin was observed over the past 3 years of study. At both centres, there was a yearly increase in resistance to ciprofloxacin, ofloxacin, amox-clav and gentamin. This result points to a lack of antimicrobial use guideline since at the PHC level, several classes of antibiotics including those that should be used in secondary and tertiary care settings such as levofloxacin and azithromycin are likely prescribed for patients. A good surveillance system is necessary for an effective antimicrobial stewardship program.

Clinical microbiology laboratories are integral component to patient care and the success of AMS by ensuring good surveillance system on antibiogram/resistogram in addition with other criteria to make good clinical decisions. In some resource-limited countries, sentinel hospitals lack even basic microbiology laboratory facilities such as lean water and uninterrupted power supply. In other regions, laboratories have some of the requisite reagents and instruments but lack skilled staff. However, clinical microbiology laboratories have not been recognized as a priority by government bodies in developing countries making it a challenge for prospective AMS activities [24, 25].

The strength of this study lies in the fact that all data available during the study period in both facilities were included, thus the results give a good impression of antimicrobial resistance in the facilities over the study period. This study is among the few available study that profiles antimicrobial resistance in primary care setting. A significant limitation of this study is the low number of samples collated from the various facilities during the study period, this limitation is further seen with few isolates for some bacteria such as proteus, *Streptococci Pneumonia* and *Neisseria gonnorhea* below the required count of 30 isolates necessary to be used for culture and sensitivity tests [26]. In addition, as a retrospective study, issues with inter-personnel variations in media preparation, procedures on culture sensitivity tests and interpretation and human error associated with manual laboratory report entry were not accounted for.

## Conclusion

In this study, low sensitivity to commonly used antimicrobials in both SHC and PHC was observed, this trend has been consistent for the past 3 years and is more in SHC. Reproductive tract and respiratory tract infections were the most represented infections caused by isolated microorganisms in this study. Multidrug resistance of isolate was generally high, higher level of MDR was observed among *S. aureus*, *pseudomonas aeruginosa* and coliform isolates. This study does not only reiterate the need for antimicrobial stewardship practice to be instituted in healthcare settings, and the role of multidisciplinary approach in use of antimicrobials in healthcare facilities, but also exposes the importance of including primary health care facilities in policies and strategies involving antimicrobial stewardship and resistance.

## Supporting information

**S1 Text. Data collection tool.**
(DOCX)

**S1 Data. Dataset.**
(SAV)

## Acknowledgments

The authors wish to appreciate staffs of Laboratories in Central hospital, Oredo Primary health care and Ugbekun Primary health care in Benin City.

## Author Contributions

**Conceptualization:** Isabel Naomi Aika, Ehijie Enato.

**Data curation:** Isabel Naomi Aika.

**Formal analysis:** Isabel Naomi Aika.

**Investigation:** Isabel Naomi Aika.

**Methodology:** Isabel Naomi Aika, Ehijie Enato.

**Project administration:** Isabel Naomi Aika, Ehijie Enato.

**Supervision:** Ehijie Enato.

**Writing – original draft:** Isabel Naomi Aika.

**Writing – review & editing:** Isabel Naomi Aika, Ehijie Enato.

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
