## [Decision Letter · Decision Letter 0]

13 Jul 2022

PGPH-D-22-00846

Antibiogram of Clinical Isolates from Primary and Secondary Healthcare Facilities: A Step Towards Antimicrobial Stewardship

Dear Dr. Aika,

Thank you for submitting your manuscript to PLOS Global Public Health. After careful consideration, we feel that it has merit but does not fully meet PLOS Global Public Health’s publication criteria as it currently stands. Therefore, we invite you to submit a revised version of the manuscript that addresses the points raised during the review process.

We look forward to receiving your revised manuscript.

Kind regards,

Ben Pascoe

Academic Editor

Journal Requirements:

1. In the Methods section and the online submission form, please provide additional information about the patient records used in your retrospective study. Specifically, please ensure that you have discussed whether all data were fully anonymized before you accessed them and/or whether the IRB or ethics committee waived the requirement for informed consent. If patients provided informed written consent to have data from their medical records used in research, please include this information.

2. Please provide separate figure files in .tif or .eps format and remove the embedded figures within the manuscript file.

Additional Editor Comments (if provided):

I have reviewed your manuscript alongside an additional reviewer, whose feedback can be found below.

This manuscript makes a valuable contribution on the characterisation and surveillance of rising AMR across the globe. Despite large scale surveillance studies in higher income countries, very little is known on the prevalence and distribution of AMR in LMICs.

Specific points:

Highlight low numbers of samples.

Additional detail in the methods on patient recruitment, bacterial identification and antibiotic susceptibility testing.

The results section needs more explanatory detail, highlighting the rationale and key findings of each analysis.

Additional detail in figure legends.

Reviewers' comments:

Reviewer's Responses to Questions

**Comments to the Author**

1. Does this manuscript meet PLOS Global Public Health’s publication criteria? Is the manuscript technically sound, and do the data support the conclusions? The manuscript must describe methodologically and ethically rigorous research with conclusions that are appropriately drawn based on the data presented.

Reviewer #1: Partly

2. Has the statistical analysis been performed appropriately and rigorously?

Reviewer #1: I don't know

3. Have the authors made all data underlying the findings in their manuscript fully available (please refer to the Data Availability Statement at the start of the manuscript PDF file)?

Reviewer #1: Yes

4. Is the manuscript presented in an intelligible fashion and written in standard English?

Reviewer #1: Yes

5. Review Comments to the Author

Reviewer #1: The manuscript is really interesting and brings unique data on antibiograms and stewardship in LMICs. However it the limited samples should be highlighted as a limitation of the study. Urine, semen and vaginal swabs might not be the main reason of consultation in PHC and particularly not in SHC , it would be good to explain a bit more why these were the more common samples rather than blood, CSF or respiratory samples.

The paper would benefit of making the link between the microbiology findings and the main syndromes/diagnosis seen at PHC and SCH; this would emphasize the need of similar studies to better define clinical guidelines for the more common syndromes (LRTI, BSI, Meningitis, etc). Very few studies show S aureus as a cause of UTI or vaginal/prostatic infections, and if this is the case in this region , it deserves a complete approach towards guidelines.

Lastly, a mention on bacteria intrinsically resistance to some antibiotics should be mention in order to avoid confusions. Some statements like high resistance to Azytromicine should be seen in the clinical context: none of the infections described (uro genital) will be treated with Azytromicine.

6. PLOS authors have the option to publish the peer review history of their article (what does this mean?). If published, this will include your full peer review and any attached files.

**Do you want your identity to be public for this peer review?** For information about this choice, including consent withdrawal, please see our Privacy Policy.

Reviewer #1: No

---

## [Editor Report · Decision Letter 1]

28 Nov 2022

Antibiogram of Clinical Isolates from Primary and Secondary Healthcare Facilities: A Step Towards Antimicrobial Stewardship

PGPH-D-22-00846R1

Dear Dr Aika,

We are pleased to inform you that your manuscript 'Antibiogram of Clinical Isolates from Primary and Secondary Healthcare Facilities: A Step Towards Antimicrobial Stewardship' has been provisionally accepted for publication in PLOS Global Public Health.

Best regards,

Ben Pascoe

Academic Editor

Thank you for taking the time to address the reviewer and editorial concerns.